# Characterising the asynchronous resurgence of common respiratory viruses following the COVID-19 pandemic

Chenkai Zhao [1], Tiantian Zhang[1], Ling Guo[1], Shiqi Sun[1], Yumeng Miao[1], Chee Fu Yung[2,3,4], Jane Tomlinson[5], Kirill Stolyarov[6], Zakhar Shchomak[7], Yong Poovorawan[8], David James Nokes [9], Carmen Muñoz-Almagro [10,11,12], Michal Mandelboim[13,14], James W. Keck[15], Joanne Marie Langley [16,17], Terho Heikkinen[18,19], Jikui Deng[20], Philippe Colson [21,22,23], Giorgi Chakhunashvili[24], Mauricio T. Caballero[25,26], Louis Bont [27,28,29], Daniel R. Feikin [30], Harish Nair [1,31], Xin Wang [31,32,39] ✉, You Li [1,31,33,39] ✉ & Respiratory Virus Global Epidemiology Network*

The COVID-19 pandemic and relevant non-pharmaceutical interventions (NPIs) interrupted the circulation of common respiratory viruses. These viruses demonstrated an unprecedented asynchronous resurgence as NPIs were relaxed. We compiled a global dataset from a systematic review, online surveillance reports and unpublished data from Respiratory Virus Global Epidemiology Network, encompassing 92 sites. We compared the resurgence timings of respiratory viruses within each site and synthesised differences in timings across sites, using a generalised linear mixed-effects model. We revealed a distinct sequential timing in the first post-pandemic resurgence: rhinovirus resurged the earliest, followed by seasonal coronavirus, parainfluenza virus, respiratory syncytial virus, adenovirus, metapneumovirus and influenza A virus, with influenza B virus exhibiting the latest resurgence. Similar sequential timing was observed in the second resurgence except influenza A virus caught up with metapneumovirus. The consistent asynchrony across geographical regions suggests that virus-specific characteristics, rather than location-specific factors, determining the relative timing of resurgence.

Human respiratory viruses such as influenza virus (IV) and respiratory syncytial virus (RSV) are important causes of severe respiratory infections[1]. Previous reports showed that most of the common respiratory viruses, including IV, RSV, parainfluenza virus (PIV), metapneumovirus (MPV), endemic seasonal coronaviruses (sCoV), and rhinovirus (RV) had distinct seasonal circulating patterns albeit with variations globally by geographical and environmental characteristics[2-4].

In 2020, the onset of the coronavirus disease 2019 (COVID-19) pandemic and the large-scale implementation of non-pharmaceutical interventions (NPIs) across the globe was associated with a precipitous and major impact on the circulation of common respiratory viruses; the activity of IV and RSV remained remarkably lower during the usual circulating season in multiple countries[5-7]. The subsequent relaxation of NPIs resulted in an unprecedented out-of-season resurgence of RSV in several countries, likely a result of depleted population immunity[8].

A full list of affiliations appears at the end of the paper. *A list of authors and their affiliations appears at the end of the paper.

✉e-mail: Xin.Wang@njmu.edu.cn; You.Li@njmu.edu.cn

However, the resurgence of these common respiratory viruses did not occur simultaneously. For example, according to the weekly sentinel surveillance report from New South Wales, Australia, in January 2022, while all respiratory viruses included in the report had substantially lower activity than the 2015–2019 average level in 2020, RV and adenovirus (AdV) appeared to be the least affected; RSV activity was suppressed until an out-of-season resurgence occurred in September 2020; by contrast, there had been limited activity of PIV and MPV until mid-2021, and circulation of IV remained at very low level by the end of 2021[9]. Interestingly, similar asynchronous resurgence was also noted in other countries with varied timing and stridency of NPIs[10,11]. It remains unclear whether the observed variations in the resurgence of these respiratory viruses are regionally specific or globally consistent, which can be crucial for enhancing future pandemic preparedness and informing public health strategies.

In this study, we compiled a large dataset on respiratory virus activity across the globe from published studies, surveillance reports, and international collaborators. We determined the onset and peak of epidemics from different data sources using a unified method. We used a generalised linear mixed-effects model (GLMM) framework to synthesise the time interval between epidemic onsets and between epidemic peaks of different respiratory viruses during the COVID-19 pandemic. (Fig. 1).

## Results

### Descriptive analysis of the study dataset

We included a total of 58 individual data sources in this study, 31 from the systematic literature search, 12 from surveillance, and 15 from unpublished research data shared by members in the Respiratory Virus Global Epidemiology Network (RSV GEN) (Fig. 2). Detailed data-source-level characteristics are available in Supplementary Tables 1–3. The compiled dataset provided data on a total of 3052 epidemics (1444 before the COVID-19 pandemic and 1608 after the onset of the pandemic) from 92 sites in 31 countries globally (Supplementary Fig. 1). Of these included sites and countries, 80 sites were from temperate regions, and 12 sites were from tropical regions; approximately half (16/31) of the countries were low- and middle-income countries. The majority of the 92 sites were rated as moderate-to-high quality (58 sites, 63%); 24% (22 sites) of the sites were rated as high quality; detailed quality assessment results can be found in Supplementary Tables 4–6.

Before the COVID-19 pandemic, viral epidemics occurred approximately every one year for IAV, sCoV, RSV, MPV and IBV, with the onset-onset intervals of two consecutive epidemics ranging from 319 days (95% CI: 292–349) for IAV to 452 days (382–535) for IBV; by comparison, RV, AdV and PIV had shorter onset-onset intervals (from 251 days [223–283] for RV to 294 days [263–330] for PIV). After the start of the COVID-19 pandemic, the onset of the first re-emergence was delayed for all respiratory viruses, as indicated by the longer onset-onset intervals; in the subsequent epidemic, the onset-onset interval became substantially shorter and returned to the pre-pandemic intervals. Same patterns were observed when using peak-peak intervals in place of onset-onset intervals (Fig. 3).

Similar findings were observed in the time intervals of epidemics between temperate and tropical regions; the only difference was in the epidemic intervals for IAV before the COVID-19 pandemic, which were shorter than one year in the tropical region (Fig. 4, Supplementary Figs. 2 and 3).

### Relative timing of the first post-pandemic resurgence

Using a matching approach, which estimated the difference in the timing of resurgence between viruses at the same site since the onset of the COVID-19 pandemic, we showed that RV resurgence earlier than all other viruses, with a lead onset of 198 to 708 days. sCoV, PIV, RSV, and AdV had similar timings of resurgence, significantly earlier than

MPV, IAV and IBV; MPV and IAV were consistently earlier than IBV (Fig. 5). Between the two IAV subtypes, H3N2 resurged significantly earlier than H1N1 by 355 days (95% CI: 299–410) and than IBV by 274 days (187–362) while the latter two had similar timings of resurgence (Supplementary Figs. 4 and 5). Among PIV subtypes, PIV-3 was the main driver of observed patterns of PIV, whereas the timings of resurgence of PIV subtypes 1, 2, and 4 were comparable to MPV and IAV (Supplementary Figs. 6 and 7).

Similar sequential order in the timing of resurgence was observed separately in the temperate and tropical regions; the only exception was that RSV, sCoV and RV (rather than RV alone) resurged earlier than other viruses in the tropical region (Fig. 6, Supplementary Figs. 8 and 9). When further stratifying the temperate regions by hemisphere, we found that while the results in the northern hemisphere resembled the global analysis, in the southern hemisphere, AdV, RSV, and RV (rather than RV alone) resurged earlier than other viruses; the confidence interval for sCoV was too wide for a meaningful comparison due to data scarcity (Supplementary Figs. 10–13). Similar results were observed when stratifying the data by country income levels (Supplementary Figs. 14–17).

All predefined sensitivity analyses consistently yielded similar results to the main analysis. (Supplementary Figs. 18–47).

### Relative timing of the second post-pandemic resurgence

Overall, the sequential order in the timing of the first resurgence persisted for the second resurgence although the magnitude of difference changed in some viruses (Fig. 5). Specifically, both IAV and IBV had shorter time lags from other viruses in the second resurgence than in the first resurgence, with the net difference between the two resurgences ranging from 26 to 118 days. By contrast, MPV had longer time lags from other viruses than IAV and IBV (the net difference ranging from 6 to 87 days), and its lead time from IAV and IBV became shorter; in fact, there were no statistical differences in the timing between MPV and IAV for the second resurgence; further analysis by IAV subtypes suggested that H1N1 mainly drove the findings of the shorter time lags for IAV in the second resurgence (Supplementary Figs. 4 and 5). Subgroup analysis by PIV subtypes showed that PIV-3 circulated relatively later in the second resurgence than in the first resurgence, despite still leading other three PIV subtypes and MPV, IAV, and IBV (Supplementary Figs. 6 and 7).

While RV, sCoV, and RSV resurged earlier than other viruses during the first resurgence in the tropical region, RV remained the earliest virus in the second resurgence, and sCoV and RSV circulated later than RV but still comparable to or earlier than other viruses in the second resurgence (Fig. 6).

## Discussion

In this study, we collected population-level aggregated data on the circulation of common respiratory viruses from multiple sources and applied a uniform analytical approach to ensure comparability across different time periods, sites and respiratory viruses. Despite the substantial variations in the calendar time of circulation of different respiratory viruses in sites with varied geographical, we identified commonalities in the patterns of circulation among these respiratory viruses. While, as expected, all viruses experienced a delayed onset after the occurrence of the COVID-19 pandemic, through matching by individual sites, we showed that the sequential order of the resurgence during the COVID-19 pandemic was virus-specific; the resurgence of RV was earliest, followed by sCoV, PIV, RSV, and AdV, subsequently by IAV and MPV, and the resurgence of IBV was the latest. Overall, consistent sequential order was observed for the second resurgence except that IAV had caught up with MPV. These findings help address key knowledge gaps regarding the circulating characteristics of common respiratory viruses in the post-pandemic era and provide insights into virus-specific factors of circulating patterns, which might include

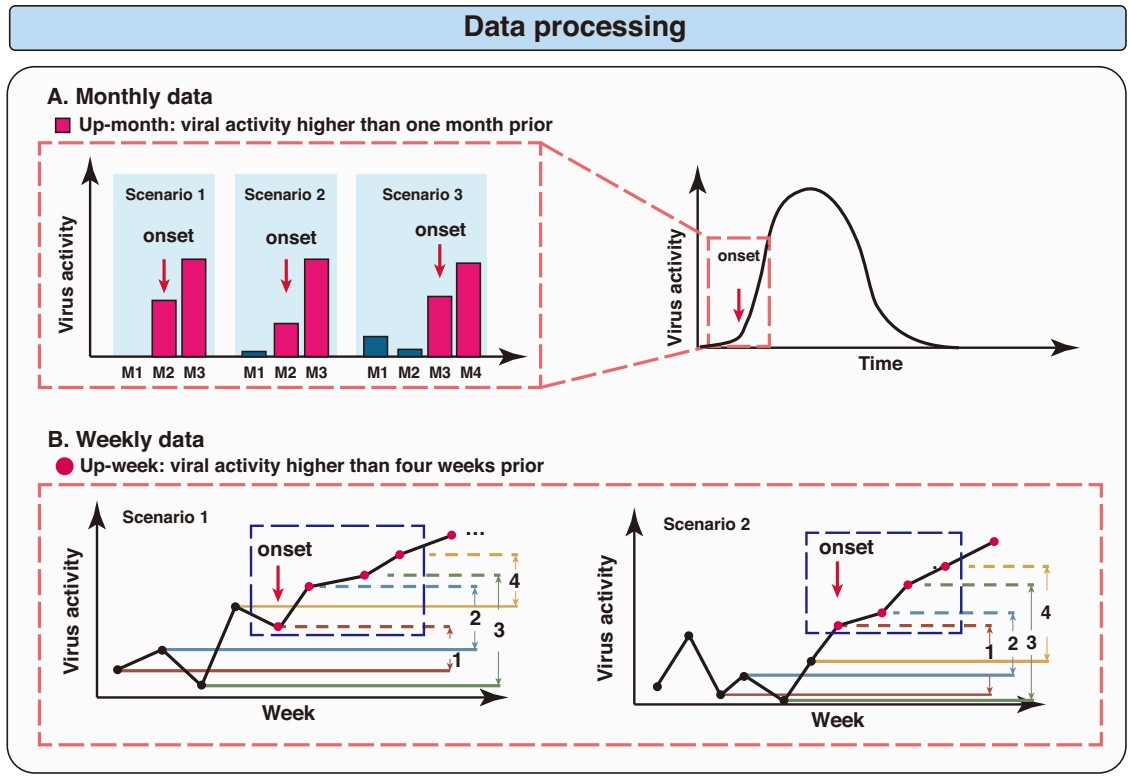

**Data processing**

**A. Monthly data**
- Up-month: viral activity higher than one month prior

**B. Weekly data**
- Up-week: viral activity higher than four weeks prior

**Data Analysis**

**Descriptive: the pooling approach**

- Calculating the time interval between two epidemics of the same virus per site, and pooled up the intervals across sites

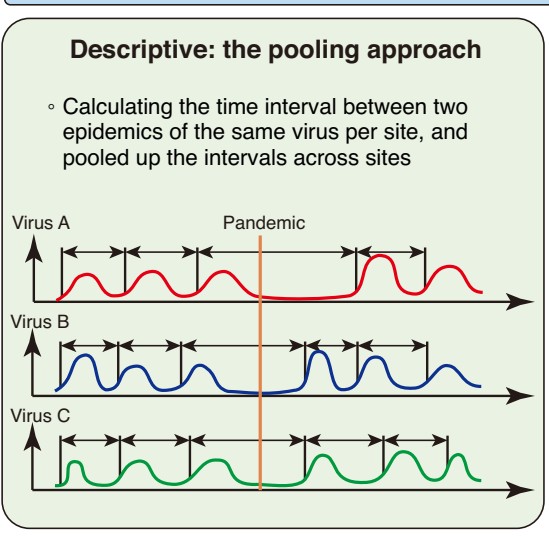

**Analytical: the matching approach**

- Calculating the differences in timing between viruses during their first and second resurgence during the pandemic, matched by site, and synthesised the differences across sites

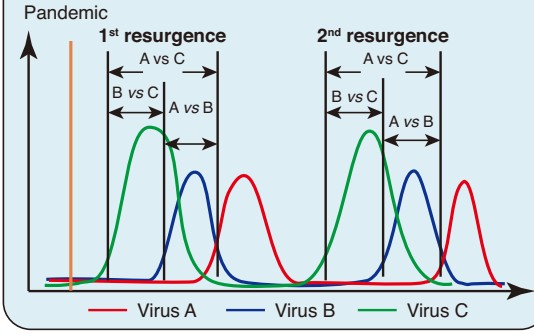

**Fig. 1 | Overview of methodology.** Data were collected from three sources: systematic literature review, online surveillance reports, and the RSV GEN network. The epidemic onset was defined using a "4-week trend" method, where each week's viral activity (case numbers or positive proportion) was compared to that of four weeks prior. An epidemic onset was identified when four consecutive weeks showed higher activity than their respective weeks four weeks earlier ("up-weeks"). Two approaches were implemented: (1) A descriptive pooling approach that compared intervals between consecutive epidemic onsets of the same virus across sites for both pre- and post-pandemic periods; (2) An analytical matching approach that examined the relative timing differences between pairs of viruses' resurgence, matched by study site.

environmental stability, infectivity, transmissibility, population immunity, and reservoirs, as well as possible virus-virus interaction.

All the respiratory viruses in this study exhibited relatively stable periodicity of circulation before the COVID-19 pandemic in both temperate and tropical regions. Unsurprisingly, following the implementation of NPIs, the resurgence of these viruses was delayed, as reported in several studies[10,12–14]. After the first re-circulation of these viruses, the periodicity of circulation for most of these viruses reverts back to the pre-pandemic period periodicity in the second resurgence.

What was interesting and not previously revealed was the distinct and consistent sequential order of resurgence of these common respiratory viruses; specifically, we showed that RV, along with RSV, sCoV, AdV, and PIV (mainly PIV-3) had relatively earlier resurgence, followed by MPV and IAV, and that IBV had the latest resurgence. Such consistency across different sites globally provides strong evidence for virus-specific transmission determinants that are independent of the common driving factors of transmission specific to the local contexts, such as the timing, duration, and intensity of NPIs, meteorological

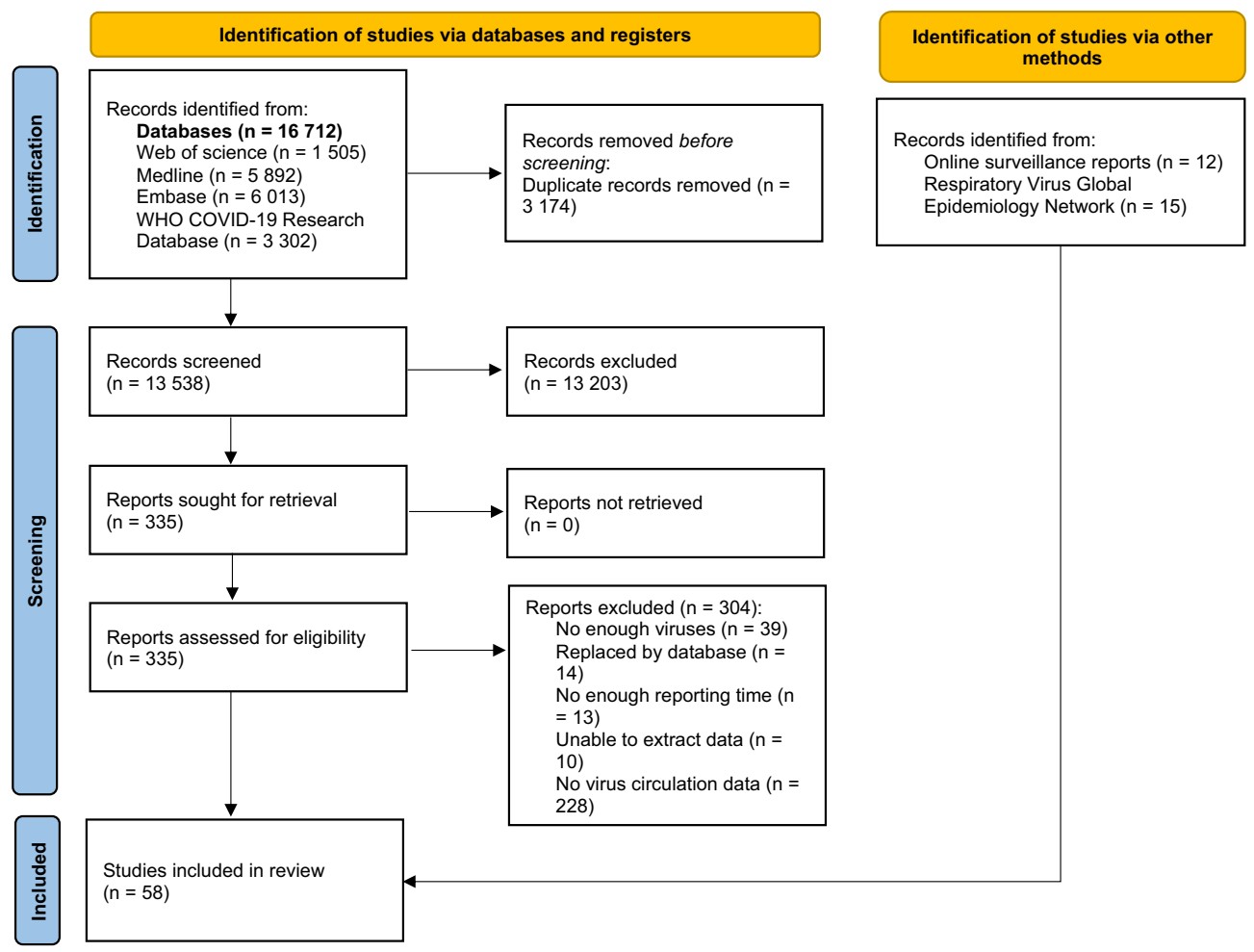

**Fig. 2 | PRISMA flow diagram.** The diagram shows the screening process of eligible data for analysis.

factors, geographical factors, COVID-19 epidemics, population mobility, behaviours and personal protection, and population vaccination history.

The following virus-specific determinants could help explain the consistency in the sequential order of the resurgence. First, the durability of immunity across different viruses could explain their varied relative timing of resurgence[15]. For example, RV has over 160 genotypes, with its observed genetic diversity, limited cross-protection and high natural mutation rates leading to high immune escape[16]. Similarly, for influenza, the antigenic regions of the hemagglutinin protein in A/H3N2 evolve more rapidly than those in A/H1N1 and IBV, which explained well why the resurgence of A/H3N2 was over 9 months earlier than other influenza (sub)types[17]. Second, preexisting community prevalence could help explain the asynchrony in resurgence. Certain viruses, such as RV and AdV, are known to be prevalent in healthy individuals and can establish asymptomatic persistence in the human respiratory tract[18,19]. These characteristics enabled RV and AdV to more easily re-establish transmission following the relaxation of NPIs or even maintain circulation despite concurrent NPIs. Wastewater-based surveillance demonstrated the correlation between viral RNA concentration with the confirmed cases[20–22], and showed that RV and AdV were continuously detected in wastewater; by contrast, influenza and RSV showed seasonal patterns of detection, and IBV was less frequently detected in wastewater[20,21,23,24]. Moreover, RV and AdV as non-enveloped viruses could persist longer on surfaces compared to enveloped viruses with lipid membranes that were sensitive to lipophilic disinfection[25]. Third, different respiratory viruses could compete for niches in the respiratory tract, with highly competitive

viruses resurging early. Virus-virus interaction was previously reported at the cellular, host, and population levels[26]. For example, In-vitro evidence showed that infection with RV could block SARS-CoV-2 replication within the respiratory epithelium[27]. During the 2009 H1N1 influenza pandemic in Europe, preceding regional RV epidemics were temporally associated with unexplained and abrupt declines in influenza cases[28,29]. A modelling study based on viral surveillance data in Scotland revealed a negative interaction between RV and IAV[30]. Fourth, the variations could be explained by the varied infectiousness of viruses. It is well appreciated that there was a trade-off between infectiousness and virulence. Viruses such as RV and AdV were more likely to have higher infectiousness and lower virulence. Previous studies showed that detection of RV or AdV in the upper respiratory tract had a less important role in the aetiology of lower respiratory infection (i.e. less virulent) than other viruses such as influenza, MPV, and RSV[19,31]. Lastly, varied transmissibility could also explain the between-virus variations. For example, viruses with aerosol transmission as the major transmission route could have higher transmissibility and were more likely to re-establish transmission following the relaxation of NPIs, such as RV[32].

We acknowledge several limitations in this study. First, similar to most global-level systematic analyses, heterogeneity could be introduced by variations in study settings (e.g. community *vs* hospital), case definitions (e.g. influenza-like illness, acute respiratory infections, severe acute respiratory infections, etc.), testing criteria, testing capacities, and testing methodologies. Nonetheless, our analysis relied only on the temporal trend of testing positive (rather than a universal threshold such as a 10% positive rate) to determine the onset and peak

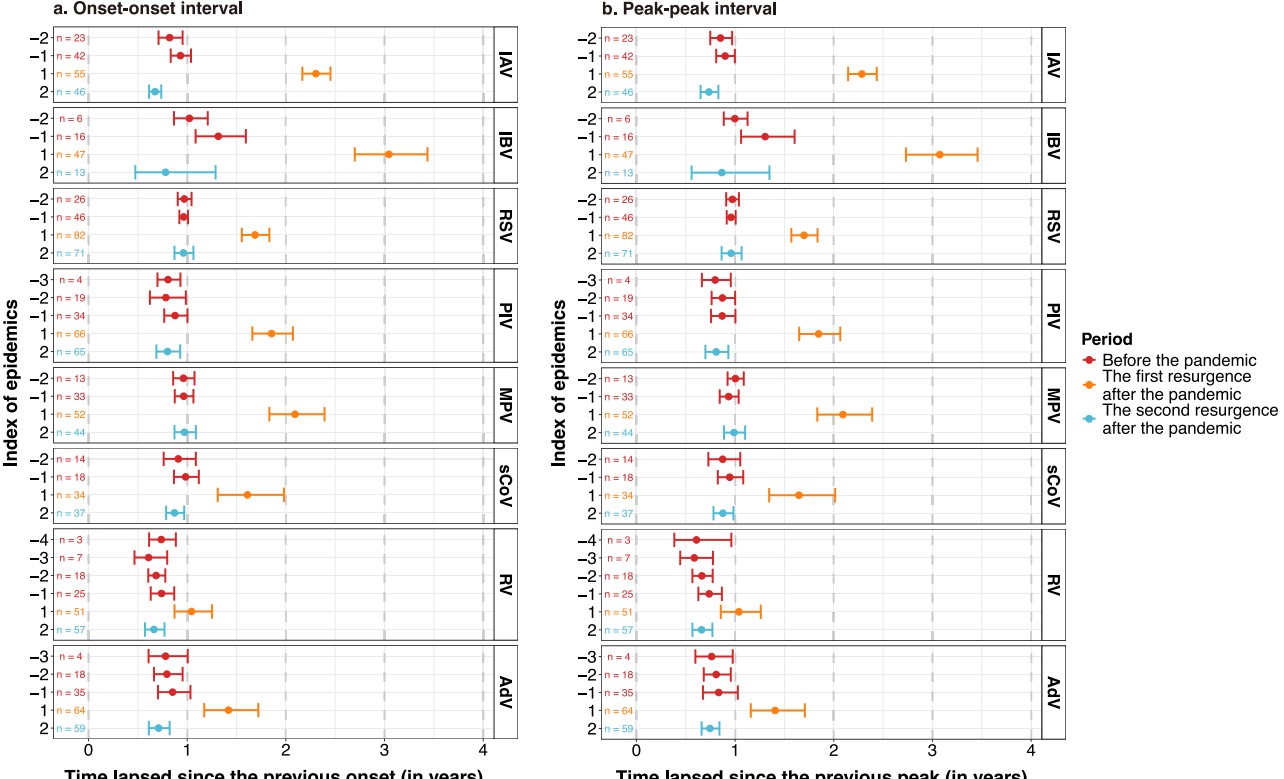

**Fig. 3 | The virus-specific time intervals between epidemics. a** Virus-specific time intervals between epidemic onsets in consecutive years. **b** Virus-specific time intervals between epidemic peaks in consecutive years. The *x*-axis represents the time elapsed since the previous onset or peak, measured in years. The left *y*-axis lists the index of epidemics relative to the onset of the COVID-19 pandemic, with negative numbers indicating epidemics occurring before the pandemic, and positive numbers indicating epidemics occurring after the onset of the pandemic. Data are presented as mean values and error bars indicate their 95% confidence intervals. IAV influenza A virus, IBV influenza B virus, RSV respiratory syncytial virus, PIV parainfluenza virus, MPV metapneumovirus, sCoV seasonal coronavirus, RV rhinovirus, AdV adenovirus. Source data are provided as a Source Data file.

of the circulation, which was insensitive to variations as mentioned above. The fact that the onset-based analysis and peak-based analysis yielded similar results further supported the robustness of our findings. Second, the data included in this study were not geographically representative and the results were driven by high-income countries (mostly in the temperate regions). To address this limitation, we conducted subgroup analyses by latitudinal group and by country income classification, which did not yield substantially different results from the global analysis. In addition, as data from different geographical scales were included, we conducted ad-hoc sensitivity analyses that demonstrated the robustness of our analysis of data at different geographical scales. Third, case-based surveillance data were prone to biases such as variations in health-seeking behaviour, reporting delays, limited diagnostic capabilities, and not covering milder infections. Fourth, different time periods were not represented equally across different study sites. Due to the time lag between study completion and publication, studies identified from the literature review did not have more recent data (e.g. in 2023). In some study sites, data on certain viruses were not available until the onset of the COVID-19 pandemic; as a result, the pre-pandemic period was less represented compared to the first two years since the onset of the pandemic. Moreover, the criteria and capacity of respiratory viral tests in a specific site could have changed over time; for example, multi-plex PCR was introduced or expanded in a number of countries during the COVID-19 pandemic. Although the scale-up of the tests could lead to increased sensitivity for detecting viral epidemics, we believe that our findings were less impacted as the differences of timings assessed in our study were matched by individual sites. Lastly, this study focused on the timing of epidemics rather than the intensity of epidemics due

to the challenges in comparing the intensity of epidemics across different viruses and study settings. Meanwhile, we were also unable to quantitatively assess the impact of NPIs due to the challenges in reconciling data on NPIs from different sites and factoring in population compliance to these NPIs.

The occurrence of the COVID-19 pandemic and the unprecedented large-scale implementation of NPIs provides a unique opportunity to observe how the transmission of different respiratory viruses was interrupted and how their transmission and circulation were later re-established. Our study reveals the changed temporal patterns of circulation as well as a virus-specific asynchrony in the timing of resurgence following the onset of the pandemic across the globe, providing insights into the complexity of circulation and transmission patterns of different respiratory viruses. These findings also emphasise the value of establishing and extending surveillance to a year-round and multi-pathogen-based system. Such characteristics could prove invaluable in preparing for and responding to potential future pandemics caused by unknown viruses.

## Methods
### Data sources
**Systematic literature review.** For the systematic literature review (PROSPERO registration: CRD42023481281), we searched four databases (Web of Science, Medline, Embase, and WHO COVID-19 Research Database) to identify studies published between 1st January 2021 and 31st December 2023 that reported viral activity data since the onset of the COVID-19 pandemic for influenza, RSV, and at least one of the following the other viruses: PIV, MPV, sCoV, RV, and AdV. We modified

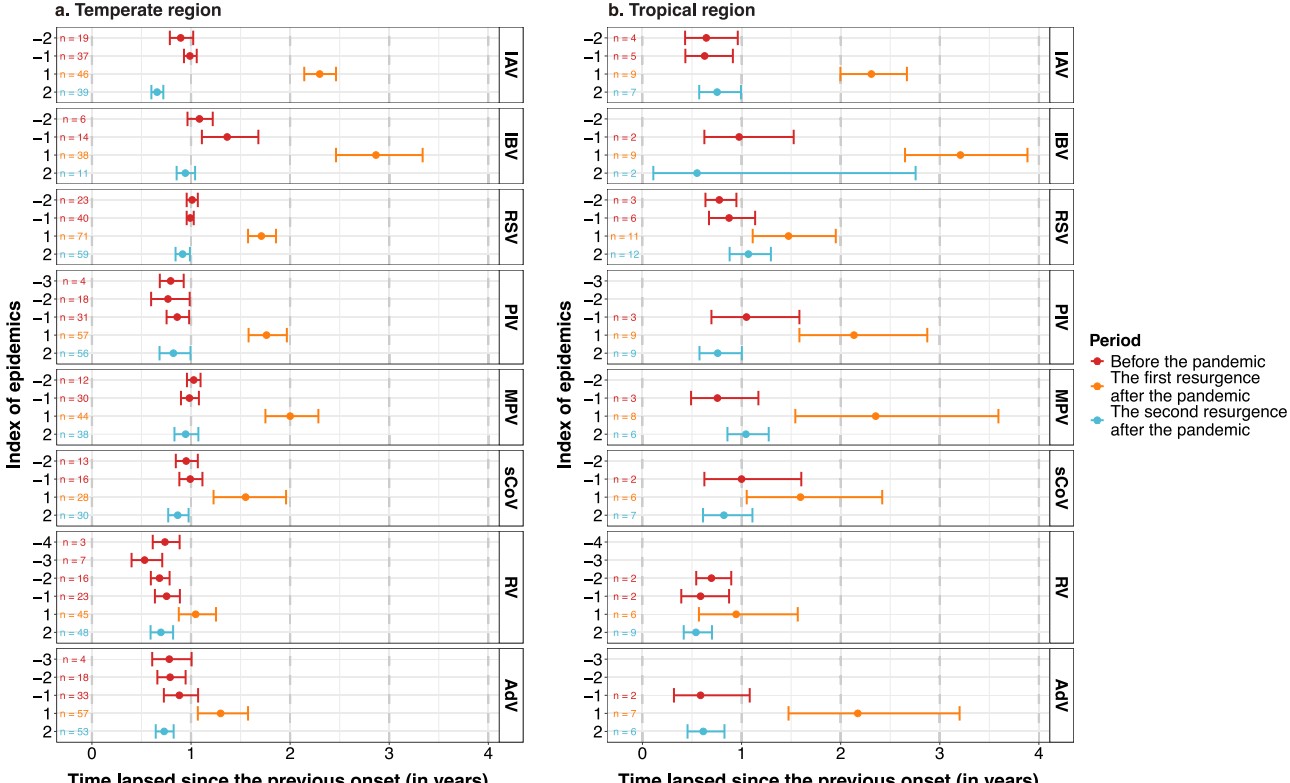

**Fig. 4 | The virus-specific time intervals between onsets by region. a** Virus-specific time intervals between epidemic onsets in consecutive years in the temperate region. **b** Virus-specific time intervals between epidemic onsets in consecutive years in the tropical region. The *x*-axis represents the time elapsed since the previous onset, measured in years. The left *y*-axis lists the index of epidemics relative to the onset of the COVID-19 pandemic, with negative numbers indicating epidemics occurring before the pandemic, and positive numbers indicating epidemics occurring after the onset of the pandemic. Data are presented as mean values and error bars indicate their 95% confidence intervals. IAV influenza A virus, IBV influenza B virus, RSV respiratory syncytial virus, PIV parainfluenza virus, MPV metapneumovirus, sCoV seasonal coronavirus, RV rhinovirus, AdV adenovirus. Source data are provided as a Source Data file.

search terms from our previously published systematic review on the seasonality of respiratory viruses[2]; these were "respiratory virus", "influenza virus", and "respiratory syncytial virus", combined with "epidemic", "prevalence", "circulate", and "surveillance" (detailed search strategy in Supplementary Methods 1). No language restrictions were applied. Eligible studies were required to report data on the number of positive cases or the positive proportion of each respiratory virus, at least monthly and covered the time period between the onset of the local COVID-19 epidemics and the end of 2020. However, there were no further restrictions with regard to regions, age groups, study designs, or diagnostic tests used (except that studies that relied on serology only were not included). Eligible studies needed to focus on the general population rather than participants with special medical conditions such as comorbidities or individuals from outbreak investigations (i.e. study participants were restricted to those in the outbreak setting, such as school).

Data extraction from published literature was completed using an Excel template. Study-level characteristics were collected, including study location, country, study period, age of study participants, clinical specimen, diagnostic tests, viruses reported, reporting frequency (e.g. weekly or monthly), and type of viral activity data (number of positives and positive proportion). The activity data for each reported virus per study were extracted from tables or figures of the publications, or from a publicly available repository when available; the online software WebPlotDigitizer (https://automeris.io/WebPlotDigitizer/) was used to help extract data from figures. Screening of studies and data extraction were conducted independently by CZ and jointly by LG and TZ, with discrepancies resolved through internal discussion within the review team.

**Online surveillance reports.** To supplement the data extracted from the systematic literature review, we accessed several online surveillance reports of respiratory viral activity, including the United States National Respiratory and Enteric Virus Surveillance System[33], Seattle Flu Alliance[34], Houston Methodist Respiratory Pathogen Epidemiology Snapshot[35], South Korea Centre for Disease Control and Prevention[36], Finnish National Infectious Diseases Register[37], Taiwan National Infectious Disease Statistics System[38], Czech Republic National institutes of health[39], Canada FluWatch[40], Pan America Health Organization FluNet[41], Weekly national flu reports of the United Kingdom[42], Scottish Health and Social Care Open Data[43], and Robert Koch Institute Influenza Weekly Reports[44]. The latest date of data retrieval was 15th April 2024.

**Respiratory Virus Global Epidemiology Network (RSV GEN).** We further reached out to members from RSV GEN to collect unpublished viral activity data[45,46]. A total of 15 study sites had available data that fulfilled the eligibility criteria, including Alaska (USA), Bangkok (Thailand), Barcelona (Spain), Buenos Aires (Argentina), Halifax (Canada), Kilifi (Kenya), Lisbon (Portugal), Marseille (France), Melbourne (Australia), Shenzhen (China), Tbilisi (Georgia), Israel, Netherlands, and Russia. A data collection template was used to collect monthly number of positive cases and positive proportion of IV, RSV, PIV, MPV, sCoV, RV, and AdV, for the duration of 1st January 2017 through 31st December 2023.

When a site had available data from more than one data source (i.e. published studies, surveillance and RSV GEN), we prioritised the inclusion of data from RSV GEN, or surveillance when RSV GEN data were not available.

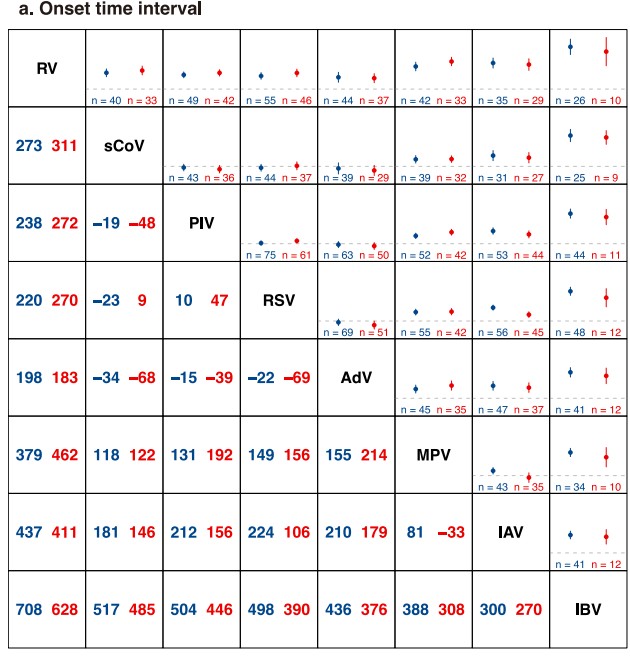

**a. Onset time interval**

**b. Peak time interval**

Fig. 5 | **The time difference of onset and peak between viruses. a** The time differences in days of onset between viruses during their first and second resurgences, matched by site. **b** The time differences in days of peak between viruses during their first and second resurgences, matched by site. The matrix presents the pairwise comparison of timing between viruses in their first (shown in blue) and second resurgence (shown in red). Positive [negative] values indicated that the virus on the top/left resurged earlier [later] than the virus on the bottom/right. Dashed line indicated no differences in the timing. Data are presented as mean values and error bars indicate their 95% confidence intervals. IAV influenza A virus, IBV influenza B virus, RSV respiratory syncytial virus, PIV parainfluenza virus, MPV metapneumovirus, sCoV seasonal coronavirus, RV rhinovirus, AdV adenovirus. Source data are provided as a Source Data file.

## Quality assessment

Quality assessment of viral activity data was conducted for all sites from data sources (a data source could report more than one site), independently by CZ and jointly by LG and TZ, with discrepancies resolved through internal discussion within the review team. A self-designed quality assessment questionnaire was used that comprised eight questions regarding representativeness, precision, and reliability (Supplementary Methods 2); the answer to each question could be "yes" (1 point), "no" (0 points), or "not clear" (0 points). Total scores could range from 0 to 8 points; sites receiving 5 points or more were considered as moderate-to-high-quality and sites receiving 7 points or more were considered as high-quality.

## Data analysis

**Defining the epidemic onset and peak.** Considering the variations in testing methods, the off-season circulation of viruses such as RSV, the likely changes in the availability and criteria for testing, and changes in behaviours for seeking care throughout the COVID-19 pandemic, the commonly used approaches for defining the epidemic onset of respiratory viruses such as the positivity threshold approach[47] and the annual average percentage[2] approach could not be properly applied. To overcome these challenges, we modified the method used for evaluating the onset of out-of-season resurgence of RSV in a previous study[8], namely the "4-week trend" method (Fig. 1). Briefly, we compared the number of cases or the positive proportion (depending on data availability) to that of four weeks earlier for each week. A week was noted as an "up-week" if the number of virus-specific cases or the positive proportion was higher than the corresponding week four weeks prior; the onset of an epidemic was defined as the beginning "up-week" of four consecutive "up-weeks". For monthly aggregated viral activity data, we adopted a similar definition, an "up-month", which was a month with higher activity than the previous month; the onset of an epidemic was defined as the first "up-month" since the absence of the last "up-month". When a data source had data on both the number of positive cases and the positive proportion, we used the positive proportion for defining the epidemic onset in the main analysis and used the number of positive cases as sensitivity analysis. Similar to the definition of onset, we defined peak as the week or month with the highest positive cases or positive proportion (the earliest peak was selected when there was more than one peak).

Considering that all data were right-censored based on a specific calendar date rather than based on the number of resurgences following the onset of the COVID-19 pandemic, to minimise potential selection bias, we focused only on up to the first two resurgences of all viruses in this study.

**Main analysis.** We used a two-step approach to understand the patterns of the virus resurgence since the COVID-19 pandemic: the pooling approach as a descriptive analysis for understanding temporal patterns of circulation of each virus, and the matching approach for analysing the relative timing of resurgence between different viruses (Fig. 1).

For the pooling approach, we calculated the time interval between the epidemic onsets in two consecutive epidemics of the same virus—the onset-onset interval for each virus and study site, and likewise the peak-peak interval. This was conducted for both the pre-pandemic and the pandemic periods when data were available. The resulting time intervals were then pooled across sites for each virus and season (treating each site as an observation), using a GLMM with a random intercept to account for heterogeneities among study sites. The results could help understand the short-term effect of the COVID-19 pandemic on the timing of epidemics.

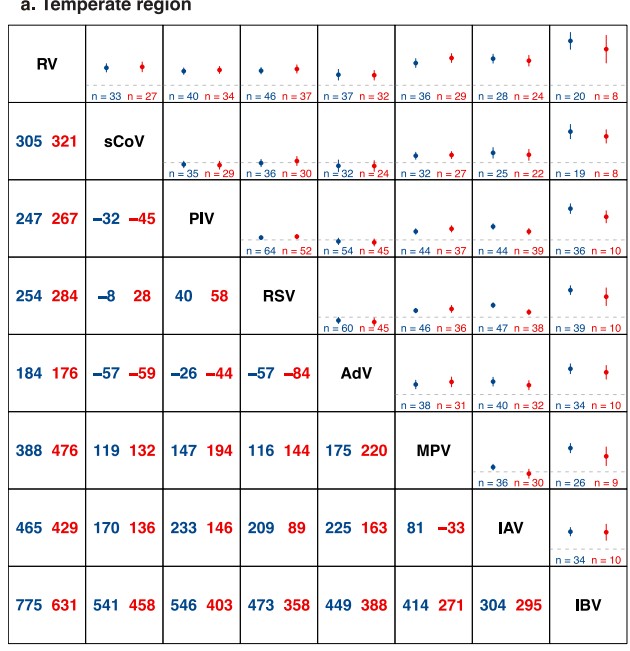

**Fig. 6 | The time difference of onset between viruses by region. a** The time differences in days of onset between viruses during their first and second resurgences, matched by sites in temperate regions. **b** The time differences in days of onset between viruses during their first and second resurgences, matched by sites in tropical regions. The matrix presents the pairwise comparison of timing between viruses in their first (shown in blue) and second resurgence (shown in red). Positive [negative] values indicated that the virus on the top/left resurged earlier [later] than the virus on the bottom/right. Dashed line indicated no differences in the timing. Data are presented as mean values and error bars indicate their 95% confidence intervals. IAV influenza A virus, IBV influenza B virus, RSV respiratory syncytial virus, PIV parainfluenza virus, MPV metapneumovirus; sCoV seasonal coronavirus, RV rhinovirus, AdV adenovirus. Source data are provided as a Source Data file.

For the matching approach, we focused on analysing the differences in the timing of the first resurgence (onset and peak) between different viruses following the COVID-19 pandemic. Specifically, we calculated the time difference between the first resurgence of a pair of viruses (all possible combinations), matched by study sites. These time differences were then pooled across sites using the same GLMM framework as in the pooling approach. This helped understand the relative differences in the timing of the first resurgence between viruses while accounting for all site-specific characteristics such as the population characteristics, timing and magnitude of COVID-19 epidemics and implementation of NPIs. We also applied the same matching approach to analyse the timing of the second resurgence of circulation for each virus. Where data were available, we repeated the analysis above to further assess influenza A virus subtypes H1N1 and H3N2, and PIV subtypes 1–4.

Two separate subgroup analyses were conducted. The first one was stratifying the analysis by up latitudinal group (temperate regions in 23.5°–66.5° latitude, which was further divided into northern and southern temperate regions, and tropical region in 23.5°N–23.5°S); the second one was stratifying by World Bank income classification (high-income, and low- and middle-income countries).

**Sensitivity analysis.** A series of predefined one-way sensitivity analyses were conducted to assess the robustness of the main analysis. Briefly, we restricted to data on positive proportion and to data on number of positives separately; restricted to data identified from the literature, and from the surveillance and RSV GEN separately; restricted to weekly aggregated data and monthly aggregated data separately; restricted to morderate-to-high-quality studies (overall quality score ≥5); and restricted to data sources that included participants of all ages.

As ad-hoc sensitivity analysis, we excluded data from Canada, Finland, and China—three countries that had most sites (potentially overrepresented)—one country at a time. In addition, for Canada and Finland, we used nationally aggregated data in place of regional aggregated data as a separate sensitivity analysis. As the included sites could have varied geographical scales, ranging from a single community or hospital to a province or even a country, we also conducted an ad-hoc exploratory analysis with the surveillance data from Canada to assess the robustness of findings when using data at different levels: national, provincial and municipal. As the approach for determining onset and peak of viral epidemics was sensitive to the sample size of positive cases in each data source, we conducted another separate analysis that included only data sources that had ≥1000 positive cases for each virus.

## Statistical software, and checklist

All data analyses and visualisations were conducted using R software (version 4.2.2). The systematic literature review in this study was reported according to the Preferred Reporting Items for Systematic Reviews and Meta-Analyses (PRISMA) guidelines.

## Reporting summary

Further information on research design is available in the Nature Portfolio Reporting Summary linked to this article.

## Data availability

All data analysed in this study (including surveillance and unpublished data) are available on GitHub (https://github.com/ChenkaiZhao-086/Vir_interaction)[48]. Source data for Figs. 3–6 are provided with this paper. Source data are provided with this paper.

## Code availability

The R codes used in this analysis are available on GitHub (https://github.com/ChenkaiZhao-086/Vir_interaction)[48].

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

## Acknowledgements

We thank the Dutch Working Group on Clinical Virology from the Dutch Society for Clinical Microbiology (NVMM) and all participating laboratories providing the Virological data from the weekly Sentinel Surveillance system. This study was supported by World Health Organisation and the National Natural Science Foundation of China (Grant no 82473692, Y.L; 82404372, X.W).

## Author contributions

Y.L. conceived the study with contributions from X.W and H.N. C.Z. led data collection with contributions from T.Z., L.G., S.S., Y.M., C.-F.Y., J.T., K.S., Z.S., Y.P., D.-J.N., C.M.-A., M.M., J.-W.K., J.-M.L., T.H., J.D., P.C., G.C., M.-T.C., and L.B. C.Z. led data analysis with inputs from Y.L. C.Z., Y.L., and X.W. co-led data interpretation, with inputs from C.Z., T.Z., L.G., S.S., Y.M., C.-F.Y., J.T., K.S., Z.S., Y.P., D.J.-N., C.M.-A., M.M., J.W.-K., J.M.-L., T.H., J.D., P.C., G.C., M.T.-C., L.B., D.-R.F., H.N., X.W., Y.L. C.Z. wrote the first draft of the report with inputs from all authors. All authors read and approved the final draft of the manuscript. C.Z. and Y.L. had full access to and verified all the study data. All authors had final responsibility for the decision to submit for publication.

## Competing interests

T.Z. reports personal fees from Pfizer, outside the submitted work. D.-J.N. reports grants from Wellcome Trust related to the submitted manuscript. C.M.-A. reports personal fees from MSD, Pfizer and Sanofi, and travel grants from MSD and Pfizer, outside the submitted work. J.-W.K. reports grants from Centers for Disease Control and Prevention (CDC) related to the submitted work, grants from National Institutes of Health, Merck Sharp and Dohme Corporation, United States Department of Agriculture, National Science Foundation, CDC and Greenwall Foundation, and participation on data safety monitoring boards for Enhancing the Diabetes Prevention Program to promote weight loss among non-responders in a community based lifestyle intervention funded by National Institute of Diabetes and Digestive and Kidney Diseases, outside the submitted work. J.-M.L. reports that her employer received grants for the conduct of clinical research from Pfizer, GSK, Sanofi, Merck, and Moderna and personal fees paid to her employer from Enanta, GSK and Sanofi. T.H. reports personal fees from MSD, Pfizer and Sanofi, and participation on data safety monitoring boards for Sanofi, Enanta, MSD, Moderna, Shionogi and Pfizer, outside the submitted work. H.N. reports grants from the Innovative Medicines Initiative, WHO, the National Institute for Health Research, Pfizer, and Icosavax; and personal fees from the Gates Foundation, Pfizer, ReViral, GSK, Merck, Icosavax, Sanofi, Novavax, and AbbVie, outside the submitted work. X.W reports grants from GSK and personal fees from Pfizer, outside the submitted work. Y.L. reports grants from WHO, the Wellcome Trust, and GSK, and personal fees from Pfizer, outside the submitted work. All other authors declare no competing interests.

## Additional information

¹Department of Epidemiology, National Vaccine Innovation Platform, School of Public Health, Nanjing Medical University, Nanjing, China. ²Infectious Diseases Service, Department of Paediatrics, KK Women's and Children's Hospital, Singapore, Singapore. ³Duke-NUS Medical School, Singapore, Singapore. ⁴Lee Kong Chian School of Medicine, Nanyang Technological University, Singapore, Singapore. ⁵Infection Prevention and Control Department, The Royal Children's Hospital Melbourne, Melbourne, Australia. ⁶Smorodintsev Research Institute of Influenza, St Petersburg, Russia. ⁷Department of Pediatrics, Hospital Santa Maria, Centro Hospitalar Universitário Lisboa Norte, Lisbon, Portugal. ⁸Center of Excellence in Clinical Virology, Faculty of Medicine, Chulalongkorn University, Bangkok, Thailand. ⁹KEMRI-Wellcome Trust Research Programme, Kilifi, Kenya. ¹⁰Institut de Recerca Sant Joan de Déu-Hospital Sant Joan de Deu, Barcelona, Spain. ¹¹CIBER Epidemiology and Public Health (CIBERESP), Madrid, Spain. ¹²Department of Medicine, Universitat Internacional de Catalunya, Barcelona, Spain. ¹³Central Virology Laboratory, Ministry of Health, Tel-Hashomer, Israel. ¹⁴Faculty of Medicine, Tel Aviv University, Tel-Aviv, Israel. ¹⁵Research Services Department, Alaska Native Tribal Health Consortium, Anchorage, AK, USA. ¹⁶Canadian Center for Vaccinology, IWK Health and Nova Scotia Health, Dalhousie University, Halifax, NS, Canada. ¹⁷Departments of Pediatrics and Community Health and Epidemiology, Dalhousie University, Halifax, NS, Canada.

[18]Department of Pediatrics, University of Turku, Turku, Finland. [19]Department of Pediatrics, Turku University Hospital, Turku, Finland. [20]Department of Infectious Diseases, Shenzhen children's hospital, Shenzhen, China. [21]Laboratory and Infectious Diseases Departments, IHU Méditerranée Infection, Marseille, France. [22]Microbes Evolution Phylogeny and Infections (MEPHI), Institut de Recherche pour le Développement (IRD), Aix-Marseille University, Marseille, France. [23]Assistance Publique-Hôpitaux de Marseille (AP-HM), Marseille, France. [24]National Center for Disease Control and Public Health, Tbilisi, Georgia. [25]Consejo Nacional de Investigaciones Científicas y Técnicas (CONICET), Buenos Aires, Argentina. [26]Centro Infant de Medicina de Traslacional (CIMeT), Escuela de Bio y Nanotecnología (EByN) Universidad Nacional de San Martín, San Martín, Buenos Aires, Argentina. [27]Department of Pediatrics, Wilhemina Children's Hospital, University Medical Center Utrecht, Utrecht, The Netherlands. [28]ReSViNET Foundation, Zeist, The Netherlands. [29]Department of Epidemiology of Microbial Diseases, Yale School of Public Health, Yale University, New Haven, CT, USA. [30]Department of Immunizations, Vaccines, and Biologicals, World Health Organization, Geneva, Switzerland. [31]Centre for Global Health, Usher Institute, University of Edinburgh, Edinburgh, UK. [32]Department of Biostatistics, National Vaccine Innovation Platform, School of Public Health, Nanjing Medical University, Nanjing, China. [33]Changzhou Third People's Hospital, Changzhou Medical Center, Nanjing Medical University, Changzhou, China. [39]These authors contributed equally: Xin Wang, You Li.
✉e-mail: Xin.Wang@njmu.edu.cn; You.Li@njmu.edu.cn

## Respiratory Virus Global Epidemiology Network

Chenkai Zhao [1], Tiantian Zhang[1], Ling Guo[1], Shiqi Sun[1], Yumeng Miao[1], Chee Fu Yung[2,3,4], Jane Tomlinson[5], Yara-Natalie Abo[34], Andrew Daley[34], Gregory Waller[35], Kirill Stolyarov[6], Daria M. Danilenko[6], Andrey B. Komissarov[6], Zakhar Shchomak[7], Teresa Bandeira[7], Maria Rosário Barreto[36], Yong Poovorawan[8], Nongruthai Suntronwong[8], Siripat Pasittungkul[8], David James Nokes [9], Esther Nyadzua Katama[9], Carmen Muñoz-Almagro [10,11,12], Alba Arranz[10], Cristian Launes[10], Quique Bassat[37], Michal Mandelboim[13,14], Ital Nemet[13], James W. Keck[15], Jennifer D. Dobson[38], Joanne Marie Langley [16,17], Terho Heikkinen[18,19], Jikui Deng[20], Jiajia Bi[20], Guangcheng Deng[20], Philippe Colson [21,22,23], Céline Boschi[21], Bernard La Scola[21], Didier Raoult[21], Giorgi Chakhunashvili[24], Irakli Karseladze[24], Khatuna Zakhashvili[24], Olgha Tarkhan-Mouravi[24], Mauricio T. Caballero[25,26], Julia Dvorkin[25], Louis Bont [27,28,29], Marie N. Billard[27], Daniel R. Feikin [30], Harish Nair [1,31], Xin Wang [31,32,39]✉ & You Li [1,31,33,39]✉

[34]Department of Microbiology and Infection Prevention and Control, The Royal Children's Hospital Melbourne, Melbourne, Australia. [35]Department of Microbiology, The Royal Children's Hospital Melbourne, Melbourne, Australia. [36]Department of Laboratory Medicine, Hospital Santa Maria, Centro Hospitalar Universitário Lisboa Norte, Lisbon, Portugal. [37]IsGlobal-Hospital Clinic, CIBER of Epidemiology and Public Health (CIBERESP), Hospital Sant Joan de Deu, University of Barcelona, Barcelona, Spain. [38]Yukon-Kuskokwim Health Corporation, Bethel, AK, USA.

