## [Transparent Peer Review file · Nature Communications]

Characterising the asynchronous resurgence of common respiratory viruses following the COVID-19 pandemic

Corresponding Author: Dr You Li

Version 0:

Reviewer comments:

Reviewer #1

(Remarks to the Author)

Dear Editor, I would like to thank for the possibility to review this important paper from a multinational research collaboration, which focused on the respiratory pathogen resurgence during the COVID-19 pandemic. The overall quality of the manuscript was fine.

Abstract was written and provided nice summary of the work. Introduction provided enough context and justification for this current work.

Methods were described in such detail that the results could be reproducible.

Results are presented clearly and the figures provide nice information which is easy to read.

Discussion: Should there be more discussion on the possibility that some part of the findings could be due to rapid increase of multiplex PCR testing in at least high income countries, which could explain that the epidemic interval is shortened as less severe cases are more frequently tested. At least that was the case in Finland, as we saw higher peak epidemics in the register of test positivity, but not that high in the clinical wards.

Overall this was very well conducted systematic analysis, which used a comprehensive strategy and provides important findings which may be used in future epidemic dynamic predictions or in preparedness processes.

(Remarks on code availability)

Reviewer #2

(Remarks to the Author)

Major comments:

The description of the results is slightly misleading. What is simply being reported is the timing between epidemics either measured through onset-onset or peak-peak interval. As such it is not surprising that timing between epidemics was decreased due to the pandemic; the pandemic suppressed other respiratory pathogen epidemics due to NPIs as the authors stated. The timing between peaks after the pandemic (as in first two epidemics after) compared to before the pandemic is the far more interesting comparison. However, it does not appear that there was anything of significance from looking at the results in Figure 3.

The authors stated: "For some viruses, such as IAV, the onset-onset intervals were even shorter than the pre-pandemic onset-onset intervals (247 days [220-276], which was 24% [14-33] shorter than the pre-pandemic interval)". However, no correction has been made for multiple selection effects present in this analysis.

(1) As data was limited to up to 2023 approximately, multiple epidemics would not have been observed in all regions and so looking at relative timing of later and later epidemic pairs would select for smaller differences in timing. For example, influenza A was allowed to consider 5 epidemics after the pandemic. The usual timing of influenza A epidemics is yearly. There was a timing of 2 years (on average) reported between the influenza A epidemic just before and just after the pandemic. So timing of 2nd-5th epidemics likely occurred in only a few sites. The sites where it occurred may just also have had noisy data and no actual epidemic occurring if robust statistical methods had been used for estimating peak or onset timing.

(2) Given most of these pathogens are seasonal it is not unreasonable to expect there to be at least a small epidemic (when no NPIs) during the winter months. Therefore, the timing between first and second epidemic following pandemic is highly dependent on if NPIs ended outside of the season or not. For example, if NPIs ended in summer we might expect an influenza A epidemic followed by a second the following winter (half a year difference). Thus, the findings are not really about the timing between epidemics, but the timing of NPIs ending.

The relative timing of the resurgence of different pathogens is a far more interesting result and less prone to biases (as each site has the same biases, but the measurement is a within site measure). This would be a much-improved manuscript if this result was highlighted more and signposted as the main result.

Additional comments

Line 90: I don't think the "four-week/ one-month circulating trend method" is a commonly known method (a simple google search returns no results). This should be removed or explained clearly in the abstract.

Line 119-122: "..., our findings suggest that the inherent characteristics of individual viruses and virus-virus interference are the likely drivers, rather than virus-host interactions, or location-specific factors such as NPIs and environmental context." Perhaps this is explained later in the main text, but nothing stated in the abstract so far clearly leads to this statement in the conclusion. For example, this was the first mention of virus-virus interference in the manuscript.

Line 137: "accumulated population susceptibility" immunity is accumulated not susceptibility. Perhaps you could write "depleted population immunity" instead.

Line 147-156: This paragraph states "It is important to determine if there are distinct patterns in the epidemiology of the resurgence of different respiratory viruses after epidemics, as it suggests the role of pathogen-specific determinants in shaping the circulating patterns of these viruses, including virus-virus interference at the host level, transmission route(s) and survival of different viruses, and virus-specific dynamics of immunity (and less of non-specific drivers such as geographical characteristics, meteorological factors, NPIs, and susceptibility of local population)." It is unclear to me how the current study is going to elucidate the role of these pathogen-specific determinants. Either the connection should be more clearly described, or this paragraph should be rewritten so that it is not overselling the contents of the manuscript. Additionally, this paragraph suggests that the analysis will determine pathogen specific roles as opposed to NPIs and susceptibility of local population; given the decrease in infection levels in 2020 was due to NPIs and this led to increased population susceptibility it is unclear how these should not be considered highly important components that are influencing the overall dynamics of pathogen resurgence.

Line 169: There can be many biases in case-based surveillance of infectious diseases that should probably be mentioned as a limitation.

Line 169: The positive proportion is not a reliable indicator for increasing epidemic activity (i.e. onset time) as increases in the positive proportion can reflect increases in the pathogen of interest or decreases in other pathogens. Did your model account for changes in the denominator (other pathogens) when estimating the onset time? There is a good preprint out on the biases in routine influenza surveillance indicators that describes these problems.

Line 183: how common were such discrepancies?

Line 196-203: In general, you have collected data from many different geographic regions. Can you justify comparing the dynamics of respiratory pathogens between countries like Russia (population >140 million across 11 time zones) with states like Alaska (population <1 million, sparsely populated), or cities like Marseilles (population <1 million, relatively densely populated)?

Methods S4 quality assessment form Q7: "for at least 5?" there needs to be a unit after 5. Probably a typo.

Methods S4 quality assessment form Q8: what do you define as a stable testing capacity?

Line 220: Figure 1 is not clearly describing the methodology to me. There is no legend explaining what is going on in the figure and it was not visually intuitive enough for me to understand. For example, in weekly data scenario A I can't see why onset is defined in the week that cases decreased? It is clearer from the text in the methods so maybe some of that text could be repeated in the legend of the figure.

Line 220-224: The definition of onset time being used is biased. Requiring the epidemic activity to have increased from four weeks prior for four weeks in a row means that pathogens that experience lower numbers of cases will be far more variable. Lower case numbers will have relatively greater noise and uncertainty in estimates for each week this means that the threshold of increases for four weeks straight may be biased.

Line 227-229: why did you preferentially use the positive proportion over the number of positive cases? As I have stated previously there are biases in both laboratory confirmed case numbers and positive proportion. I think that the positive proportion is one of the worst indicators to use so it would be useful to justify this decision with a reference to something stating that it is better. Lots of papers use the positive proportion without justifying why it is a good indicator, and there has been recent work suggesting it is the most biased indicator used for influenza surveillance.

Line 240: "... virus to help understand the impact of the COVID-19 pandemic on the periodicity of viral circulation" Explicitly state the short term effect on timing of epidemics. Your results will likely not generalise to patterns over the coming decade. Similarly periodicity suggests that the epidemic occur at a constant period was this the case? It is more likely that resurgences were seen with relaxation of NPIs followed by a few irregular years. These epidemics may now be back to a periodic pattern but using the term "periodicity" for those short term dynamics is misleading.

Line 253-257: did you consider latitudinal group and World Bank income level simultaneously or separately? Looking at table S3 it seems that a lot of regions were duplicated. For example you included data for multiple provinces in Ontario Canada, and then also province of Ontario in its entirety. Additionally, how robust are your estimates when some countries/regions are more highly represented than others. For example, from my count at least 21 sites are in Canada (out of 98 sites total), and at least 14 sites are in China. Figure S1 shows visually the problem with the data being used. Line 295: "epidemics of respiratory viruses reported in each study are available in the appendix (pp 42-89)" There is no description in the appendix of what is being plotted in these graphs? Also is there a repository with all the data somewhere so that it can be checked how reproducible the findings are?

(Remarks on code availability)

I was not able to check if the code would run as there were no instructions on how to read the data in. The instructions stated all analysis could be run from the 'main.R' script, but the code called some sort of data object called 'all_Vir' that had not been defined anywhere in the script. Also I think the package dplyr was missing from the list of packages to be called.

Reviewer #3

(Remarks to the Author)

This paper used published and unpublished data on respiratory pathogens to describe the patterns of respiratory diseases both pre- and post-COVID-19 pandemic. The authors took a comprehensive approach, with a robust sensitivity analysis, to assess the intervals between epidemics and then try an infer patterns by virus. The patterns, which have been largely documented by others at the time they were occurring, are consistent with what we know and understand about the different viruses. Rhinoviruses reemerged first, then RSV then influenza viruses. These data point to virus-specific differences that are critical to understand our best approaches going forward in the arena of prevention and control, and specifically preparedness. Adding more of that context in the discussion could be most helpful to the readers.

Specific comments

Line 151-152. It is not immediately clear why the resurgence patterns would be less driven by NPIs and susceptibility. Please explain why you hypothesize that, as it has great implications for how you interpret your findings. (Also, you say susceptibility is a big factor in line 373 which contradict this original statement).

Line 168-171. Please address what diagnostic methods were included. Were these only molecular detections or was it broader? I see you capture that data on line 177; however, if this was not limited in some way it could be problematic for comparison reasons. Here I am specifically thinking of serologic detections.

Line 213 or 230. In the analytic approach did you restrict to a minimum number of specimens tested per week for a certain number of weeks of the year? This would be important since some surveillance systems collapsed during the pandemic and having a minimum requirement might at least ensure better quality data.

Line 215. I think you should specifically add changes in behaviors for seeking care.

Line 272. While having a high number cut-off for the total positive specimens is good, I think it is perhaps more critical that the total number of specimens tested and the collection approach we uniform each week. Was that explored?

Line 295-296. I am very surprised that 58/92 were of high quality. I was unable to find the appendices to review the scale more closely. Most surveillance data is inherently flawed. It is still useful for analyses such as the one presented here, but give the limitations outlined in the discussion it seems that the quality might score lower. I suppose you could have high internal validity but much heterogeneity across sites which limits the pooling approach.

Line 361. Why is susceptibility or population immunity not in this list? If fact you include it in line 373.

Line 368-369. Please clarify what you mean by, "...seemingly to offset the delay in the first re-circulation?"

Line 439-441. If you could not assess the impact of NPIs then I am not sure how you can state in the introduction (lines 151-152) that they are not a driver of the patterns seen in your paper. If you cannot assess then you cannot know.

Line 448-450. I wonder if you can contrast what you are saying (that viruses have unique characteristics and we are better served by not lumping all together), with the current trend for pan-respiratory disease prevention and response. I think your findings have big implications and somewhat contradict the current global trend. This discussion could really make your paper relevant to current preparedness discussions.

(Remarks on code availability)

Version 1:

Reviewer comments:

Reviewer #2

(Remarks to the Author)

The authors have addressed all of my comments with detailed and thoughtful responses. I think the manuscript is much improved and have no comments that would warrant changes to the paper at this time.

My only comment (which requires no changes to the document) is in response to their response to my third comment discussing potential problems using their method for defining onset and peak timing. The authors stated: "As shown in the figure below using data from the Netherlands, our approach, despite being less sophisticated, could effectively filter out noise in the time series of viral activity ..."

This suggests that the authors can determine what is noise and what is not in the signal. A statistical methodology would define and quantify the noise in a way that is universally interpretable. Different statistical methods would determine different patterns in signals/noise but would be consistent. The method used is effectively a simple statistical method that only works when the signal:noise ratio is high. The figure used to demonstrate the robustness of the model if anything makes me more concerned. The 'timing' of the second RSV epidemic does not look well estimated, and all of the RV epidemics determined do not look accurate. This is of course my interpretation of the signal/noise which might not match yours. In the future the authors should consider more robust approaches for consistently quantifying noise --- for example, running a simple model that assumes smooth trends and quantifies the noise around the smoothed trend.

(Remarks on code availability)

Reviewer #3

(Remarks to the Author)

The revised paper does a very good job at addressing the reviewer comments and I have no further suggestions.

(Remarks on code availability)

Reviewer comments

Authors' note: for ease of referencing, we have renumbered the comments.

Reviewer #1 (Remarks to the Author):

Comment #1

Dear Editor, I would like to thank you for the possibility to review this important paper from a multinational research collaboration, which focused on the respiratory pathogen resurgence during the COVID-19 pandemic. The overall quality of the manuscript was fine.

Abstract was written and provided nice summary of the work. Introduction provided enough context and justification for this current work.

Methods were described in such detail that the results could be reproducible.

Results are presented clearly and the figures provide nice information which is easy to read.

Discussion: Should there be more discussion on the possibility that some part of the findings could be due to rapid increase of multiplex PCR testing in at least high income countries, which could explain that the epidemic interval is shortened as less severe cases are more frequently tested. At least that was the case in Finland, as we saw higher peak epidemics in the register of test positivity, but not that high in the clinical wards.

Overall this was very well conducted systematic analysis, which used a comprehensive strategy and provides important findings which may be used in future epidemic dynamic predictions or in preparedness processes.

Response:

Thank you for the overall comment on our study and for the specific comment regarding increase of PCR testing in some countries.

We agree that the increased PCR testing capacity and the related change in criteria for testing could lead to higher sensitivity in detecting viral epidemics / onsets. This affects the results from the pooling approach, but less on the matching approach. As suggested by the editor and other reviewers, we have now considered the pooling approach only as a descriptive approach and considered matching approach as the only analytical approach that our conclusion is drawn upon.

Nonetheless, we agree that the changes over time in the testing capacity and criteria are important and have added the following statements in the limitation of the manuscript, as follows –

"Moreover, the criteria and capacity of respiratory viral tests in a specific site could have changed over time; for example, multi-plex PCR was introduced or expanded in a number of

countries during the COVID-19 pandemic. Although the scale-up of the tests could lead to increased sensitivity for detecting viral epidemics, we believe that our findings were less impacted as the differences of timings assessed in our study were matched by individual sites."

Changes made to the manuscript:

Lines 265-269, Pages 11-12.

Reviewer #2 (Remarks to the Author):

Comment #2

Major comments:

The description of the results is slightly misleading. What is simply being reported is the timing between epidemics either measured through onset-onset or peak-peak interval. As such it is not surprising that timing between epidemics was decreased due to the pandemic; the pandemic suppressed other respiratory pathogen epidemics due to NPIs as the authors stated. The timing between peaks after the pandemic (as in first two epidemics after) compared to before the pandemic is the far more interesting comparison. However, it does not appear that there was anything of significance from looking at the results in Figure 3. The authors stated: "For some viruses, such as IAV, the onset-onset intervals were even shorter than the pre-pandemic onset-onset intervals (247 days [220-276], which was 24% [14-33] shorter than the pre-pandemic interval)". However, no correction has been made for multiple selection effects present in this analysis.

Response:

We appreciate the comment.

We would like to clarify that the focus of our analysis was less on the absolute delay in resurgence (we agree that the delay itself is unsurprising) and more on the relative delay between viruses and more importantly the specific sequential order of the resurgence.

Therefore, as also suggested by the *editorial comment* above and **Comment #4** below, we have now considered the reported onset-onset and peak-peak intervals from the pooling approach only as a descriptive analysis, and considered the matching approach as the main approach / model that our conclusion is drawn solely upon. We have also revised throughout the manuscript to help clarify the focus of the study.

Additionally, we have removed the quantitative comparison of onset-onset intervals between two epidemics of the same virus (i.e., from the pooling approach) as it was highly impacted by the NPI itself.

Comment #3

(1) As data was limited to up to 2023 approximately, multiple epidemics would not have been observed in all regions and so looking at relative timing of later and later epidemic pairs would select for smaller differences in timing. For example, influenza A was allowed to consider 5 epidemics after the pandemic. The usual timing of influenza A epidemics is yearly. There was a timing of 2 years (on average) reported between the influenza A

epidemic just before and just after the pandemic. So timing of 2nd-5th epidemics likely occurred in only a few sites. The sites where it occurred may just also have had noisy data and no actual epidemic occurring if robust statistical methods had been used for estimating peak or onset timing.

Response:

Thank you for raising this important point regarding the temporal limitations of our dataset. We have carefully reviewed the dataset and found that the dataset could well capture the first two resurgences but less on the third resurgence and beyond. For example, among 61 sites that provided data on influenza A virus up to the end of 2023, 53 sites (87%) observed two resurgences.

Therefore, we have decided to restrict the analysis to the first two resurgences; note that for matching approach, we have added the analysis for the second resurgence. This has now been clarified in the method section, as follows –

“Considering that all data were right-censored based on a specific calendar date rather than based on the number of resurgences following the onset of the COVID-19 pandemic, to minimise potential selection bias, we focused only on up to the first two resurgences of all viruses in this study.”

Moreover, we would like to respond to the comment regarding the robustness of our approach for determining onset and peak. First, there are no methods for estimating peak or onset timings that can be considered the “gold standard”; in fact, some of the commonly used methods could not work well during the COVID-19 pandemic since there were likely rapid changes in testing practice. This is why we used the alternative approach that relied on the temporal trend for determining onset / peak timings, which was less sensitive to changes in testing practice and reporting variations in multiple data sources.

As shown in the figure below using data from the Netherlands, our approach, despite being less sophisticated, could effectively filter out noise in the time series of viral activity (the first dot denotes onset and the second denotes peak), even for rhinovirus that demonstrated year-round-like circulation.

Changes made to the manuscript:

Lines 365-368, Page 15.

Comment #4

(2) Given most of these pathogens are seasonal it is not unreasonable to expect there to be at least a small epidemic (when no NPIs) during the winter months. Therefore, the timing between first and second epidemic following pandemic is highly dependent on if NPIs ended outside of the season or not. For example, if NPIs ended in summer we might expect an influenza A epidemic followed by a second the following winter (half a year difference). Thus, the findings are not really about the timing between epidemics, but the timing of NPIs ending.

The relative timing of the resurgence of different pathogens is a far more interesting result and less prone to biases (as each site has the same biases, but the measurement is a within site measure). This would be a much-improved manuscript if this result was highlighted more and signposted as the main result.

Response:

Thank you for the valuable suggestion. As mentioned in **Comment #2**, we agreed that the pooling approach that assessed the time intervals between epidemics of the same virus was highly dependent on NPI and was less relevant to the focus of the study, which was to understand the asynchrony of resurgence among different respiratory viruses.

By comparison, as the reviewer correctly pointed out, the matching approach was more

robust and less prone to biases, including the impact of NPIs. Therefore, we have restructured our analysis by considering the pooling approach as only a descriptive analysis and the matching approach as the main analytical approach that was used to draw conclusions. For the matching approach, we have also added the analysis for the second resurgence since the onset of the COVID-19 pandemic.

In the manuscript, we have clarified the roles of the pooling approach and matching approach in the method section, as follows –

“We used a two-step approach to understand the patterns of the virus resurgence since the COVID-19 pandemic: the pooling approach as a descriptive analysis for understanding temporal patterns of circulation of each virus, and the matching approach for analysing the relative timing of resurgence between different viruses. (Figure 1)”

Changes made to the manuscript:

Lines 370-373, Page 16.

Additional comments

Comment #5

Line 90: I don't think the “four-week/ one-month circulating trend method” is a commonly known method (a simple google search returns no results). This should be removed or explained clearly in the abstract.

Response:

Thank you for the comment. During the revision phase, we are required by the journal to reduce the word count in the abstract to no more than 150 words. Therefore, the name of the method has been removed.

Changes made to the manuscript:

The abstract.

Comment #6

Line 119-122: “..., our findings suggest that the inherent characteristics of individual viruses and virus-virus interference are the likely drivers, rather than virus-host interactions, or location-specific factors such as NPIs and environmental context.” Perhaps this is explained later in the main text, but nothing stated in the abstract so far clearly leads to this statement in the conclusion. For example, this was the first mention of virus-virus interference in the manuscript.

Response:

Thank you for the comment. We have removed the mention of virus-virus interference and revised the sentence as follows –

“The consistently distinct asynchrony across geographical regions suggests the role of virus-specific characteristics, rather than location-specific factors in determining the relative timing of resurgence.”

Changes made to the manuscript:

Lines 81-83, Page 4.

Comment #7

Line 137: “accumulated population susceptibility” immunity is accumulated not susceptibility. Perhaps you could write “depleted population immunity” instead.

Response:

Thank you for the good suggestion. We have now replaced “accumulated population susceptibility” with “depleted population immunity”.

Changes made to the manuscript:

Lines 98, Page 5.

Comment #8

Line 147-156: This paragraph states “It is important to determine if there are distinct patterns in the epidemiology of the resurgence of different respiratory viruses after epidemics, as it suggests the role of pathogen-specific determinants in shaping the circulating patterns of these viruses, including virus-virus interference at the host level, transmission route(s) and survival of different viruses, and virus-specific dynamics of immunity (and less of non-specific drivers such as geographical characteristics, meteorological factors, NPIs, and susceptibility of local population).” It is unclear to me how the current study is going to elucidate the role of these pathogen-specific determinants. Either the connection should be more clearly described, or this paragraph should be rewritten so that it is not overselling the contents of the manuscript. Additionally, this paragraph suggests that the analysis will determine pathogen specific roles as opposed to NPIs and susceptibility of local population; given the decrease in infection levels in 2020 was due to NPIs and this led to increased population susceptibility it is unclear how these should not be considered highly important components that are influencing the overall dynamics of pathogen resurgence.

Response:

Thank you. We agree with the comment and have removed the speculation about the roles

of pathogen-specific and non-pathogen-specific factors in driving the resurgence. This paragraph has now been rewritten as follows –

“In 2020, the onset of the coronavirus disease 2019 (COVID-19) pandemic and the large-scale implementation of non-pharmaceutical interventions (NPIs) across the globe was associated with a precipitous and major impact on the circulation of common respiratory viruses; the activity of IV and RSV remained remarkably lower during the usual circulating season in multiple countries⁵⁻⁷. The subsequent relaxation of NPIs resulted in an unprecedented out-of-season resurgence of RSV in several countries, likely a result of depleted population immunity⁸. However, the resurgence of these common respiratory viruses did not occur simultaneously. For example, according to the weekly sentinel surveillance report from New South Wales, Australia, in January 2022, while all respiratory viruses included in the report had substantially lower activity than the 2015–2019 average level in 2020, RV and adenovirus (AdV) appeared to be the least affected; RSV activity was suppressed until an out-of-season resurgence occurred in September 2020; by contrast, there had been limited activity of PIV and MPV until mid-2021, and circulation of IV remained at very low level by the end of 2021⁹. Interestingly, similar asynchronous resurgence was also noted in other countries with varied timing and stridency of NPIs^{10,11}. It remains unclear whether the observed variations in the resurgence of these respiratory viruses are regionally specific or globally consistent, which can be crucial for enhancing future pandemic preparedness and informing public health strategies.”

Changes made to the manuscript:

Lines 93-109, Page 5.

Comment #9

Line 169: There can be many biases in case-based surveillance of infectious diseases that should probably be mentioned as a limitation.

Response:

Thank you for the comment. We agree that case-based surveillance can have many biases. We added some discussion on this limitation, as follow–

“Third, case-based surveillance data were prone to biases such as variations in health-seeking behaviour, reporting delays, limited diagnostic capabilities, and not covering milder infections.”

Changes made to the manuscript:

Lines 258-259, Page 11.

Comment #10

Line 169: The positive proportion is not a reliable indicator for increasing epidemic activity (i.e. onset time) as increases in the positive proportion can reflect increases in the pathogen of interest or decreases in other pathogens. Did your model account for changes in the denominator (other pathogens) when estimating the onset time? There is a good preprint out on the biases in routine influenza surveillance indicators that describes these problems.

Response:

Thank you for the comment and for sharing the preprint (which we believe has been published at: <https://onlinelibrary.wiley.com/doi/10.1111/irv.70050>).

We agree that the positive proportion could be influenced by other co-circulating viruses. The choice between using confirmed case counts versus positivity proportions in surveillance data presents inherent trade-offs, as systematically compared in our previous study (<https://www.eurosurveillance.org/content/10.2807/1560-7917.ES.2024.29.5.2300244?crawler=true>). For the purpose of defining onset / peak of an epidemic, as pointed out by the reviewer, positivity proportions could be affected by the presence of other circulating viruses while the number of absolute counts is not without its limitation as it is sensitive to the changes in testing practice.

In this study, we did not consider one indicator superior to the other indicator. For studies that reported confirmed case counts and positivity proportions simultaneously, we determined a priori that we used the positive proportion as the primary indicator for our analysis and used the other one in sensitivity analysis, which showed consistent findings with the main analysis (Figure 5 vs Supplementary Figures 18-21). This suggests that the use of either positive proportion or absolute case count did not practically affect the findings of this analysis.

Comment #11

Line 183: how common were such discrepancies?

Response:

We appreciate the comment. These discrepancies were overall uncommon but could happen when the data were extracted from figures with relatively low resolution from published papers (using a figure recognition software named WebPlotDigitizer). We addressed this limitation by giving a lower quality assessment score (to the fifth question) in our quality assessment (details in Supplementary Methods 2); data sources with low quality assessment score (<5) were excluded as sensitivity analysis.

Comment #12

Line 196-203: In general, you have collected data from many different geographic regions. Can you justify comparing the dynamics of respiratory pathogens between countries like Russia (population>140 million across 11 time zones) with states like Alaska (population<1 million, sparsely populated), or cities like Marseilles (population <1 million, relatively densely populated)?

Response:

We appreciate the comment. While we admit that the data included in this study were from different geographical regions with varied population density among other contexts, the focus of our analysis was to demonstrate a consistently distinct relative timing of resurgence of different common respiratory viruses despite variations in the site-specific context. Therefore, it was precisely our aim to include as many sites with variable characteristics as possible to test the robustness of our findings. This also enabled us to further investigate different latitudinal groups (such as temperate region vs tropical region).

For the case of Russia, although we admit that the data crosses multiple time zones, different zones are relatively on the same range of latitudes. Our previously published global analysis of seasonality of four respiratory viruses (<https://pubmed.ncbi.nlm.nih.gov/31303294/>) suggested that sites with similar latitudes tended to have similar seasonal patterns.

Regarding the inclusion of data from varied geographical levels, we have conducted an ad-hoc sensitivity analysis using data at national, provincial, and municipal levels of Canada to verify the robustness of our findings. As shown below, for both pooling and matching approaches, the results of different geographical levels were generally similar despite varied widths of confidence intervals. In our study, we prioritised the inclusion of sites with smaller geographical scales if data from different levels of geographical scales were available.

In the method section, the following has been added –

“As the included sites could have varied geographical scales, ranging from a single community or hospital to a province or even a country, we also conducted an ad-hoc exploratory analysis with the surveillance data from Canada to assess the robustness of findings when using data at different levels: national, provincial and municipal.”

Changes made to the manuscript:

Lines 408-411, Page 17; Supplementary Figure 42-45.

Comment #13

Methods S4 quality assessment form Q7: “for at least 5?” there needs to be a unit after 5. Probably a typo.

Response:

Thank you for your comment. We revised the Q7 as follows–

“Whether viral activity data were available for at least five types of viruses?”

Changes made to the manuscript:

Supplementary Methods 2

Comment #14

Methods S4 quality assessment form Q8: what do you define as a stable testing capacity?

Response:

We appreciate the comment. We evaluated testing stability by two distinct criteria. First, whether the data source had any documented major interruptions in testing during the pandemic. Second, whether the number of specimens taken was consistently zero for more than two weeks.

We have added these criteria as a footnote to Supplementary Methods 2 to ensure clarity in our methodology.

Changes made to the manuscript:

Supplementary Methods 2

Comment #15

Line 220: Figure 1 is not clearly describing the methodology to me. There is no legend explaining what is going on in the figure and it was not visually intuitive enough for me to understand. For example, in weekly data scenario A I can't see why onset is defined in the week that cases decreased? It is clearer from the text in the methods so maybe some of that text could be repeated in the legend of the figure.

Response:

Thank you for the suggestion. We have added more explanatory texts to Figure 1 to explain "up-week" and "up-month" and added more figure legends for better clarity.

Regarding the first scenario for the weekly data, what we compared to for each week was the four weeks prior rather than the previous week. As that week had higher activity than four weeks ago (despite lower activity than one week ago), the week was determined as an "up-week". Furthermore, as the week was the first "up-week" of the four consecutive "up-weeks", the week was then determined as the onset.

Changes made to the manuscript:

Figure 1.

Comment #16

Line 220-224: The definition of onset time being used is biased. Requiring the epidemic activity to have increased from four weeks prior for four weeks in a row means that pathogens that experience lower numbers of cases will be far more variable. Lower case numbers will have relatively greater noise and uncertainty in estimates for each week this

means that the threshold of increases for four weeks straight may be biased.

Response:

Thank you for the comment. As mentioned in the response to **Comment #3** our approach could effectively filter out noise in the time series of viral activity. Nonetheless, we agree that our approach tended to be less stable when the absolute number of cases was low (although we argue that this was the common challenge for determining timing of viral epidemics).

Despite the issue of small sample size, given that the relative differences in the onset and in the peak were similar, indicating that the approach for defining onset was relatively robust. Sample size was one of the quality assessment criteria in our study (the fourth question in Supplementary Method 4); our sensitivity analysis that excluded low-quality data sources (total scores <5) yielded similar findings to the main analysis (Supplementary Figure 30-31).

Moreover, we conducted a more straightforward ad-hoc sensitivity analysis that excluded all data sources with the number of positive cases less than 1000 for any viruses. This analysis also yielded similar findings to the main analysis (Supplementary Figures 46-47). The analysis was described as follows in the method section–

“As the approach for determining onset and peak of viral epidemics was sensitive to the sample size of positive cases in each data source, we conducted another separate analysis that included only data sources that had ≥ 1000 positive cases for each virus.”

Changes made to the manuscript:

Lines 411-414, Page 17; Supplementary Figure 46-47.

Comment #17

Line 227-229: why did you preferentially use the positive proportion over the number of positive cases? As I have stated previously there are biases in both laboratory confirmed case numbers and positive proportion. I think that the positive proportion is one of the worst indicators to use so it would be useful to justify this decision with a reference to something stating that it is better. Lots of papers use the positive proportion without justifying why it is a good indicator, and there has been recent work suggesting it is the most biased indicator used for influenza surveillance.

Response:

We appreciate the comment. As discussed in the response to **Comment #10**, we agreed that both indicators had inherent biases, and we essentially did not prefer one to the other in our analysis. As not all data sources reported both indicators, we included whichever

indicator was available in individual sources.

Nonetheless, for data sources that reported both indicators, we had to make a rather arbitrary decision a priori that we prioritised the reported positive proportion in the main analysis and the absolute case counts in the sensitivity analysis, which showed consistent findings (Supplementary Figure 18-21).

Comment #18

Line 240: "... virus to help understand the impact of the COVID-19 pandemic on the periodicity of viral circulation" Explicitly state the short term effect on timing of epidemics. Your results will likely not generalise to patterns over the coming decade. Similarly periodicity suggests that the epidemic occur at a constant period was this the case? It is more likely that resurgences were seen with relaxation of NPIs followed by a few irregular years. These epidemics may now be back to a periodic pattern but using the term "periodicity" for those short term dynamics is misleading.

Response:

We appreciate the comment and have revised the sentence as follows –

"The results could help understand the short-term effect of the COVID-19 pandemic on the timing of epidemics."

Changes made to the manuscript:

Lines 380-381, Page 16.

Comment #19

Line 253-257: did you consider latitudinal group and World Bank income level simultaneously or separately?

Response:

We appreciate the comment. We considered the latitudinal group and World Bank income level separately. To avoid any ambiguity and ensure clarity in methods, we revised the sentence as follows–

"Two separate subgroup analyses were conducted. The first one was stratifying the analysis by up latitudinal group (temperate regions in 23.5°–66.5° latitude, which was further divided into northern and southern temperate regions, and tropical region in 23.5°N-23.5°S); the second one was stratifying by World Bank income classification (high-income, and low- and middle-income countries)."

Changes made to the manuscript:

Lines 393-397, Page 16-17.

Comment #20

Looking at table S3 it seems that a lot of regions were duplicated. For example you included data for multiple provinces in Ontario Canada, and then also province of Ontario in its entirety. Additionally, how robust are your estimates when some countries/regions are more highly represented than others. For example, from my count at least 21 sites are in Canada (out of 98 sites total), and at least 14 sites are in China. Figure S1 shows visually the problem with the data being used.

Response:

Thank you for your comment.

First, when a site / region had available data from more than one data source (i.e., published studies, surveillance and RSV GEN), we prioritised the inclusion of data from RSV GEN, or surveillance when RSV GEN data were not available. This has now been clarified in the method section, as follows –

“When a site had available data from more than one data source (i.e., published studies, surveillance and RSV GEN), we prioritised the inclusion of data from RSV GEN, or surveillance when RSV GEN data were not available.”

Regarding the case of Ontario, Canada, we admit that this was duplicate and have removed the duplicated data.

Moreover, we admitted that some countries were more represented than others such as China, Finland, and Canada. Therefore, we have conducted additional ad-hoc sensitivity analyses that removed the data from one of these countries at a time; for Canada and Finland, as national-level data were also available (despite it not being included in the main analysis), we replaced the site-specific data with national-level data in Canada and Finland as another ad-hoc sensitivity analysis. All of these sensitivity analyses yielded similar results to the main analysis, testifying the robustness of our findings. The detailed description of this ad-hoc sensitivity analysis was added to the method section, as follows –

“As ad-hoc sensitivity analysis, we excluded data from Canada, Finland, and China—three countries that had most sites (potentially overrepresented)—one country at a time. In addition, for Canada and Finland, we used nationally aggregated data in place of regional aggregated data as a separate sensitivity analysis.”

Changes made to the manuscript:

Lines 332-334, Page 14; Lines 405-408, Page 17; Supplementary Figures 34-41.

Comment #21

Line 295: “epidemics of respiratory viruses reported in each study are available in the appendix (pp 42-89)” There is no description in the appendix of what is being plotted in these graphs? Also is there a repository with all the data somewhere so that it can be checked how reproducible the findings are?

Response:

We appreciate the comment. We meant to present the results of onset and peak for each included site but now realised that there were too many data to present in the appendix. Alternatively, we have now provided these data in the form of a dataset in GitHub (https://github.com/ChenkaiZhao-086/Vir_interaction/blob/main/Data/Full-text%20table-V1.8.1.xlsx), which can also be used to reproduce the study findings (**Comment #22**)

Reviewer #2 (Remarks on code availability):

Comment #22

I was not able to check if the code would run as there were no instructions on how to read the data in. The instructions stated all analysis could be run from the 'main.R' script, but the code called some sort of data object called 'all_Vir' that had not been defined anywhere in the script. Also I think the package dplyr was missing from the list of packages to be called.

Response:

Thank you for the comment.

We have restructured the code and tested the code across macOS (M2 Max chip) and MS Windows platforms. To ensure successful implementation, please note these technical requirements: 1. Rtools must be installed. 2. The packages glmmTMB, TMB and Matrix need to be installed from source. 3. We recommend checking your package versions against those specified in our documentation to ensure compatibility. Hopefully, you can access and run the code this time.

In the final report, we used the results obtained from the macOS platform. It is worth noting that the results on MS Windows showed minor differences.

Regarding the package dependencies, since tidyverse inherently includes dplyr, there is no need for a separate dplyr call in the code.

Reviewer #3 (Remarks to the Author):

Comment #23

This paper used published and unpublished data on respiratory pathogens to describe the patterns of respiratory diseases both pre- and post-COVID-19 pandemic. The authors took a comprehensive approach, with a robust sensitivity analysis, to assess the intervals between epidemics and then try to infer patterns by virus. The patterns, which have been largely documented by others at the time they were occurring, are consistent with what we know and understand about the different viruses. Rhinoviruses reemerged first, then RSV then influenza viruses. These data point to virus-specific differences that are critical to understand our best approaches going forward in the arena of prevention and control, and specifically preparedness. Adding more of that context in the discussion could be most helpful to the readers.

Response:

We greatly appreciate your review. Based on your suggestions, we have revised the discussion section and rewritten the implications of our study. Please refer to our point-by-point response below for details.

Specific comments

Comment #24

Line 151-152. It is not immediately clear why the resurgence patterns would be less driven by NPIs and susceptibility. Please explain why you hypothesize that, as it has great implications for how you interpret your findings. (Also, you say susceptibility is a big factor in line 373 which contradicts this original statement).

Response:

Thank you for the comment. We agree that both NPIs and increased susceptibility play important roles in resurgences, although they could not fully explain why we observed the asynchrony in the resurgence among different viruses. Nevertheless, we do realize that the original statement could be misleading. In this study, as we did not aim to fully investigate the roles of different factors (NPIs, susceptibility, etc.), we have decided to revise the text and not discuss the roles of these factors.

As also suggested by **Reviewer #2** (in **Comment #8**), we have now revised this sentence, as follows –

“...It remains unclear whether the observed variations in the resurgence of these respiratory viruses are regionally specific or globally consistent, which can be crucial for enhancing future pandemic preparedness and informing public health strategies.”

Changes made to the manuscript:

Lines 107-109, Page 5.

Comment #25

Line 168-171. Please address what diagnostic methods were included. Were these only molecular detections or was it broader? I see you capture that data on line 177; however, if this was not limited in some way it could be problematic for comparison reasons. Here I am specifically thinking of serologic detections.

Response:

Thank you for the comment.

Serologic studies were not included. In our study, the majority of included studies were based on PCR testing, with a small proportion using antigen testing or a combination of PCR and antigen testing. We have clarified this point in the Methods section as follows–

“However, there were no further restrictions with regard to regions, age groups, study designs, or diagnostic tests used (except that studies that relied on serology only were not included).”

Changes made to the manuscript:

Lines 299-300, Page 13.

Comment #26

Line 213 or 230. In the analytic approach did you restrict to a minimum number of specimens tested per week for a certain number of weeks of the year? This would be important since some surveillance systems collapsed during the pandemic and having a minimum requirement might at least ensure better quality data.

Response:

We greatly appreciate the comment and agree that it is important to make sure that there were sufficient specimens for detecting viral epidemics over the study period.

In this study, although we did not have restrictions in terms of the number of specimens testing per week (because not all studies reported this indicator), we had a specific quality assessment question to assess the stability of testing for each data source (Q8, Supplementary Method 2). This was based on two criteria: whether the data source had any documented major interruptions in testing during the pandemic; and whether the number of specimens taken was consistently zero for more than two weeks.

Moreover, we had a threshold regarding the total number of positive cases for each reported virus (1000 cases) as part of our quality assessment (Q4, Supplementary Method 2). There was also an ad-hoc sensitivity analysis that included only data sources with ≥ 1000 cases for each virus.

Comment #27

Line 215. I think you should specifically add changes in behaviors for seeking care.

Response:

Thank you for the good suggestion. Now added.

Changes made to the manuscript:

Lines 347-348, Page 15.

Comment #28

Line 272. While having a high number cut-off for the total positive specimens is good, I think it is perhaps more critical that the total number of specimens tested and the collection approach we uniform each week. Was that explored?

Response:

Thank you for your comment. As discussed in the response to **Comment #26**, we were unable to assess the number of specimens for all data sources as such data were not available in some of these data sources. The best we could do was to use what was reported to help understand the stability of testing practice.

We admit that the criteria and capacity of respiratory viral tests in a specific site could have changed over time and have now discussed this as a limitation of this study, as follows –

“Moreover, the criteria and capacity of respiratory viral tests in a specific site could have changed over time; for example, multi-plex PCR was introduced or expanded in a number of countries during the COVID-19 pandemic. Although the scale-up of the tests could lead to increased sensitivity for detecting viral epidemics, we believe that our findings were less impacted as the differences of timings assessed in our study were matched by individual sites.”

Changes made to the manuscript:

Lines 265-269, Pages 11-12.

Comment #29

Line 295-296. I am very surprised that 58/92 were of high quality. I was unable to find the appendices to review the scale more closely. Most surveillance data is inherently flawed. It is still useful for analyses such as the one presented here, but give the limitations outlined in the discussion it seems that the quality might score lower. I suppose you could have high internal validity but much heterogeneity across sites which limits the pooling approach.

Response:

We appreciate the comment. Generally, we agree with the reviewer regarding the overall quality of surveillance data and we would like to clarify the quality assessment scale and how we defined high quality.

The full assessment scale can be found in Supplementary Method 4, consisting of 8 questions, with each being rated 1 point (indicating good quality) or 0 points; detailed assessment results for individual sites can be found in Supplementary Tables 4-6. The total score could range from 0 to 8. Our original threshold for high-quality data was ≥ 5 points (58 / 92 studies could be rated as high quality by this threshold). However, we realised that the set threshold was a bit lower and have now revised the threshold as follows (in the methods section)–

“...sites receiving 5 points or more were considered as moderate-to-high-quality and sites receiving 7 points or more were considered as high-quality.”

As shown in the figure below, the quality score ranged from 1 to 7 among the included data sources; there were 22 sites (24%) that were considered as high quality by the updated threshold.

Lastly, we agreed that compared to the matching approach that controlled for site-specific characteristics, the pooling approach was more prone to biases. Following the suggestion from the editor and **Reviewer #2**, we have restructured our analysis by considering the pooling approach as only a descriptive analysis and the matching approach as the main analytical approach that was used to draw conclusions (refer to **Comments #2 and #4**).

Changes made to the manuscript:

Lines 342-343, Page 14.

Comment #30

Line 361. Why is susceptibility or population immunity not in this list? If fact you include it in line 373.

Response:

Thank you for the comment. We added the “population immunity” to the sentence.

Changes made to the manuscript:

Lines 195, Page 9.

Comment #31

Line 368-369. Please clarify what you mean by, “...seemingly to offset the delay in the first re-circulation?”

Response:

Thank you for the comment. This was part of the discussion of the results from the pooling approach, which showed that the onset-onset intervals for some of the viruses during the subsequent epidemics (following the first resurgence) were even shorter than their pre-pandemic intervals. However, as now we have considered the pooling approach only as a descriptive analysis and restricted the analysis to only the first two resurgences (details in **Comment #3**), we have removed this discussion from the main text.

Comment #32

Line 439-441. If you could not assess the impact of NPIs then I am not sure how you can state in the introduction (lines 151-152) that they are not a driver of the patterns seen in your paper. If you cannot assess then you cannot know.

Response:

Thank you for the comment. We agree that assessing the roles of NPIs among other factors was beyond the scope of this study. Therefore, as also suggested by **Reviewer #2 (Comment #8)** and mentioned in **Comment #24**, we have removed the statement regarding NPIs in the introduction section, and re-written the corresponding paragraph as follows –

“In 2020, the onset of the coronavirus disease 2019 (COVID-19) pandemic and the large-scale implementation of non-pharmaceutical interventions (NPIs) across the globe was associated with a precipitous and major impact on the circulation of common respiratory viruses; the activity of IV and RSV remained remarkably lower during the usual circulating season in multiple countries⁵⁻⁷. The subsequent relaxation of NPIs resulted in an unprecedented out-of-season resurgence of RSV in several countries, likely a result of depleted population immunity⁸. However, the resurgence of these common respiratory viruses did not occur simultaneously. For example, according to the weekly sentinel

surveillance report from New South Wales, Australia, in January 2022, while all respiratory viruses included in the report had substantially lower activity than the 2015–2019 average level in 2020, RV and adenovirus (AdV) appeared to be the least affected; RSV activity was suppressed until an out-of-season resurgence occurred in September 2020; by contrast, there had been limited activity of PIV and MPV until mid-2021, and circulation of IV remained at very low level by the end of 2021⁹. Interestingly, similar asynchronous resurgence was also noted in other countries with varied timing and stringency of NPIs^{10,11}. It remains unclear whether the observed variations in the resurgence of these respiratory viruses are regionally specific or globally consistent, which can be crucial for enhancing future pandemic preparedness and informing public health strategies.”

Changes made to the manuscript:

Lines 107-109, Page 5.

Comment #33

Line 448-450. I wonder if you can contrast what you are saying (that viruses have unique characteristics and we are better served by not lumping all together), with the current trend for pan-respiratory disease prevention and response. I think your findings have big implications and somewhat contradict the current global trend. This discussion could really make your paper relevant to current preparedness discussions.

Response:

We appreciate the suggestion and would like to clarify regarding the main message of this study.

As different respiratory viruses have unique circulating characteristics, we cannot use the circulation data of one to represent another. This means that we will need to conduct a pan-respiratory pathogen surveillance system rather than, for example, an influenza-only surveillance system. As multi-plex PCR testing has become more available and affordable, a year-round and multi-pathogen-based system is what we would like to recommend. We have revised the corresponding sentence as follows –

“These findings also emphasise the value of establishing and extending surveillance to a year-round and multi-pathogen-based system.”

Changes made to the manuscript:

Lines 281-282, Page 12.

Reviewer comments

Reviewer #2 (Remarks to the Author):

The authors have addressed all of my comments with detailed and thoughtful responses. I think the manuscript is much improved and have no comments that would warrant changes to the paper at this time.

My only comment (which requires no changes to the document) is in response to their response to my third comment discussing potential problems using their method for defining onset and peak timing. The authors stated: "As shown in the figure below using data from the Netherlands, our approach, despite being less sophisticated, could effectively filter out noise in the time series of viral activity ..."

This suggests that the authors can determine what is noise and what is not in the signal. A statistical methodology would define and quantify the noise in a way that is universally interpretable. Different statistical methods would determine different patterns in signals/noise but would be consistent. The method used is effectively a simple statistical method that only works when the signal:noise ratio is high. The figure used to demonstrate the robustness of the model if anything makes me more concerned. The 'timing' of the second RSV epidemic does not look well estimated, and all of the RV epidemics determined do not look accurate. This is of course my interpretation of the signal/noise which might not match yours. In the future the authors should consider more robust approaches for consistently quantifying noise --- for example, running a simple model that assumes smooth trends and quantifies the noise around the smoothed trend.

Response:

Thank you very much for your encouraging feedback and for recognizing the improvements we have made to our manuscript. We greatly appreciate the time and effort you have taken to assess our work.

Regarding your insightful comments on our methodology for defining onset and peak timing, we acknowledge the limitations you pointed out concerning the distinction between signal and noise, especially in scenarios with lower signal-to-noise ratios (typically in year-round circulated viruses like RV and AdV or out-of-season circulated viruses like RSV circulated twice in one year in some country after the COVID-19 pandemic). Your suggestion to adopt more robust statistical approaches for consistently quantifying noise is highly valuable. In future research, we plan to explore and integrate such methodologies to enhance the reliability and interpretability of our analyses. We are grateful for your constructive feedback, which will undoubtedly help us refine our research further.

Reviewer #3 (Remarks to the Author):

The revised paper does a very good job at addressing the reviewer comments and I have no further suggestions.

Response:

Thank you very much for your positive feedback and for recognizing the improvements we have made to our manuscript. We appreciate your time and effort in reviewing our work.